# Characteristics and Treatment of Primary Hepatic Perivascular Epithelioid Cell Tumor (PEComa) in Adults: A Systematic Review

**DOI:** 10.3390/cancers17142276

**Published:** 2025-07-08

**Authors:** Konstantinos Papantoniou, Ioanna Aggeletopoulou, Maria Kalafateli, Christos Triantos

**Affiliations:** 1Division of Gastroenterology, Department of Internal Medicine, University of Patras, 26504 Patras, Greece; med6162@ac.upatras.gr (K.P.); chtriantos@upatras.gr (C.T.); 2Division of Digestive Diseases, Department of Metabolism, Digestion and Reproduction, Imperial College London, London W2 1NY, UK; m.kalafateli@imperial.ac.uk

**Keywords:** liver, perivascular epithelioid cell tumors, PEComa, adults, systematic review

## Abstract

Perivascular epithelioid cell tumors (PEComas) are a rare group of neoplasms characterized by their presence in areas surrounding blood vessels and the expression of special immunohistochemical markers. Reported cases regarding PEComas found in the liver are extremely rare, and specific guidelines regarding decision-making for these patients are currently lacking. The lack of distinctive clinical and radiological characteristics makes differential diagnosis with other tumors of the liver a challenge for clinicians. In this systematic review, we aim to characterize the clinical presentation, diagnostic approach, treatment, and outcomes of adult patients with hepatic PEComa. We hope that the information we provide will assist clinicians in their everyday practice and improve patient care, while also giving directions for future prospective studies on this subject.

## 1. Introduction

Perivascular epithelioid cell tumors (PEComas) are a rare group of mesenchymal neoplasms that were originally described in 1992 [1]. The unique features of these tumors are mainly attributed to their composition. Perivascular epithelioid cells are a type of cell with distinctive histological and immunohistochemical characteristics. They are typically located in areas surrounding blood vessels, while they also express both melanocytic and smooth muscle markers [2]. Angiomyolipomas (AMLs), lymphangioleiomyomatosis, clear-cell “sugar” tumors, and tumors classified as PEComas not otherwise specified (PEComa—NOS) are classified as members of this group of neoplasms [3,4]. These tumors may arise in different locations including the female reproductive tract, skin, bones, lungs, gastrointestinal (GI) tract, and kidneys. Their prevalence appears to be higher in females [5]. Most PEComas are benign; however, malignant cases have also been reported [6,7].

The liver is a rare location for the development of PEComas. Some of these tumors, especially AMLs, may appear due to tuberous sclerosis complex (TSC), a genetic disorder characterized by the presence of tumors located in different body parts and a distinct set of neurodevelopmental syndromes known as TSC-associated neurological disorders [8]. However, many liver PEComas are sporadic. Surgical excision of these neoplasms has proven to be effective, but the lack of distinctive radiological and clinical features makes preoperative diagnosis challenging. As primary liver PEComas are rare, most knowledge regarding their clinical course and treatment is derived from single case reports and case series. The lack of specific guidelines regarding the clinical approach to these tumors only adds to the difficulties that clinicians face while treating these patients [9]. These neoplasms also often resemble other liver abnormalities, such as hepatocellular carcinoma (HCC) and focal nodular hyperplasia (FNH), and inappropriate treatment might be chosen [10].

A systematic search and identification of cases of primary hepatic PEComas could consolidate data from different reports and give a more comprehensive picture of clinical characteristics and established or emerging treatment modalities directed against these neoplasms. The aim of the present study was to systematically review reported case reports and case series of primary liver PEComas to determine patient baseline characteristics, clinical course, pathological, immunohistochemical, and radiological tumor features, the method and timing of diagnosis, and treatment types. These findings could help establish a standardized diagnostic and therapeutic approach for these patients.

## 2. Materials and Methods

### 2.1. Search Strategy

A systematic literature search in PubMed, Scopus, and Cochrane libraries was conducted on 25 May 2025 to retrieve all relevant articles. This article is a systematic review written in accordanceto the guidelines from the Preferred Reporting Items for Systematic Reviews and Meta-analyses (PRISMA) [11]. The following search terms were used: ((perivascular epithelioid cell tumor) OR (PEComa)) AND ((liver) OR (hepatic)).

### 2.2. Study Selection

EndNote version 21 was used to delete duplicate files. The titles and abstracts of the unique articles were screened first independently by two different authors (K.P. and I.A.). Full-text screening was then performed again by two independent authors (K.P. and I.A.), and all potentially eligible cases of primary liver PEComa were included. The references for each article were manually assessed to complete the research.

### 2.3. Eligibility Criteria

In this systematic review, we included case series and case reports of adult patients who were diagnosed with liver PEComa after the age of 18. Articles were included only if they were written in the English language and if they provided information regarding the patient’s initial symptoms and follow-up of at least 2 months after diagnosis. In order to reduce potential limitations in data interpretation, studies providing only aggregated data or mean values, without individual patient-level information, were not included. Cases that did not fulfill these criteria and other non-relevant article types, such as reviews and expert opinions, were excluded. Studies that focused only on specific tumor characteristics, such as radiological findings or immunohistochemistry, were also excluded. Regarding articles based on the same patient dataset, papers with the longest follow-up or the largest sample size were included. Disagreements between the authors regarding study eligibility were resolved by consensus. If no agreement could be reached, conflicts were resolved after discussion with a third senior author (C.T.).

### 2.4. Data Extraction and Synthesis

For each study, the following information were extracted based on a pre-specified form: first author, year of publication, continent, age, sex, comorbidities, diagnosis of tuberous scleroris, symptoms at presentation, concurrent primary PEComa in a different organ, liver tumor size, tumor nodules, tumor location on the liver, method of diagnosis, pre-surgical biopsy, PEComa cell type, malignancy, metastasis, surgery, arterial chemoembolization, adjuvant treatment, neoadjuvant treatment, immunohistochemical markers, follow-up duration, and relapse. We used the data from all available cases to create a secondary cohort. References were constructed via EndNote 21. Version 29.0.2.0 of IBM SPSS for Windows was used for statistical analysis.

## 3. Results

We identified a total of 987 records from the three databases, along with an additional 20 records through manual reference checks (Figure 1). After removing 137 duplicates, 870 articles remained for title and abstract screening. Subsequently, 390 full-text articles were evaluated for eligibility. Ultimately, 145 studies reporting on 281 patients with primary liver PEComa were included in the analysis. Individual study origins, comorbidities, imaging modalities, immunohistochemical profiles, and follow-up details are provided in Appendix A.

In Table 1, we provide a comprehensive overview of patient demographics, clinical presentation, diagnostic workup, tumor characteristics, and treatment modalities for the primary hepatic PEComa cases included in the current study.

The majority of studies originated from Asia (97 studies, 66.6%), followed by Europe (35 studies, 24.1%), North America (8 studies, 5.5%), Africa (3 studies, 2.1%), South America (1 study, 0.7%), and Australia (1 study, 0.7%). Of the total, 48 studies (33.1%) were from China, and 24 (16.55%) were from Japan. Most studies were single case reports (121, 83.4%), while 24 (16.6%) were case series. The largest case series included was published by Jung et al. [12] and Jiang et al. [13], involving 23 and 19 patients, respectively.

The mean age of patients was 46 years (IQR: 35.25–53.75). The majority (208 patients, 74%) were female, while 73 (26%) were male. Most cases (197, 70.1%) had no significant medical history. Reported liver comorbidities included hepatitis B (23 patients, 8.2%), hepatitis C (5 patients, 1.8%), metabolic dysfunction-associated steatotic liver disease (MASLD) (3 patients, 1.1%), HCC (3 patients, 1.1%), hemangioma (3 patients, 1.1%), and cirrhosis (3 patients, 1.1%). Additionally, nine patients (3.2%) were diagnosed with tuberous sclerosis, and six (2.1%) were diagnosed with Li–Fraumeni syndrome. A history of extrahepatic malignancy was noted in 30 patients (10.7%).

The most commonly reported symptoms were abdominal pain (84 patients, 29.9%), abdominal discomfort (24 patients, 8.54%), fever (15 patients, 5.34%), weight loss (10 patients, 3.56%), fatigue (7 patients, 2.49%), vomiting (5 patients, 1.78%), and nausea (5 patients, 1.78%). Elevated alpha-fetoprotein (AFP) levels were noted only in three patients (1.1%). PEComa was detected incidentally in 157 patients (55.9%) who were asymptomatic. In 26 patients (9.3%), the tumor was discovered as a palpable mass. It was identified through abdominal ultrasound in 120 patients (43.7%), CT in 43 patients (13.7%), MRI in 11 patients (3.9%), and during unrelated abdominal surgery in 2 cases. Methods for initial tumor detection were not described for 79 patients.

During the diagnostic workup, 152 patients (54.1%) underwent ultrasound, 221 (78.6%) had CT scans, 156 (55.5%) underwent MRI, and 31 (11%) had PET-CT imaging. Tumor enhancement in the arterial phase after contrast administration with subsequent washout sign in the delayed or portal phases was observed in 93 cases (33.1%). Radiological diagnosis correctly identified PEComa in only 29 cases (10.3%), while differential diagnoses often included HCC, adenoma, focal nodular hyperplasia, and hemangioma.

Preoperative biopsy was performed in 59 patients (21%), with 44 (74.5%) receiving a confirmed PEComa diagnosis. The median tumor size was 5.15 cm (IQR: 3–9.3), with the largest tumor, measuring 30 × 25 cm, reported by Zhou et al. [14]. A single liver lesion was present in 249 patients (88.6%), and multiple nodules were present in 22 (7.8%) cases, while data were missing for 10 patients.

Tumor location was predominantly in the right hepatic lobe (149 cases, 53%), followed by the left lobe (101 cases, 33.9%) and the caudate lobe (12 cases, 4.3%). In 13 cases (4.6%), tumors extended across both the right and left lobes, with 1 case (0.4%) involving both the right and caudate lobes, and another (0.4%) involving the left and caudate lobes. Localization was not specified in four cases.

Immunohistochemical analysis was conducted in 268 cases (95.4%). Tumors were most frequently positive for HMB-45 (261/264, 98.9%), MELAN-A (124/132, 93.2%), smooth muscle actin (SMA) (179/200, 89.5%), and vimentin (62/79, 78.5%). They were commonly negative for S-100 (85/126, 67.5%), desmin (45/83, 54.2%), cytokeratins (123/125, 98.1%), epithelial membrane antigen (EMA) (26/28, 92.9%), and HepPar-1 (43/44, 97.7%).

Final diagnosis was established through radiology in 29 cases (10.3%), biopsy in 44 cases (15.7%), and post-surgical histopathology in 208 cases (74%). The tumors were classified as angiomyolipoma (AML) in 197 patients (70.1%), PEComa-NOS in 74 (26.3%), CCMMT of the falciform ligament or ligamentum teres in 7 (2.5%), and clear-cell sugar tumor in 3 (1.1%). Concurrent PEComas were found in the kidney (14 cases, 5%), lungs (1 case, 0.4%), and both kidneys and omentum (1 case, 0.4%).

Surgery was the primary treatment in 251 patients (89.3%), with 1 receiving neoadjuvant therapy and 3 undergoing adjuvant treatment postoperatively. An initial non-surgical approach was taken in 30 patients (10.7%), of whom 13 (43.3%) eventually required surgery due to tumor progression. Additional treatments included transarterial chemoembolization (TACE) in 11 patients (3.9%), transarterial embolization (TAE) in 3 (1.1%), radiofrequency ablation (RFA) in 3 (1.1%), sorafenib in 3 (1.1%), tamoxifen in 1 (0.4%), microwave ablation in 1 (0.4%), and liver transplantation in 1 patient (0.4%).

A malignant phenotype was reported in 30 cases (10.7%), with major vessel invasion observed in 8 (2.85%). One patient died before discharge following the initial hospitalization. Among those discharged, the median follow-up duration was 24 months (IQR: 12–48). Recurrence occurred in 17 patients (6%), and 7 patients (2.5%) died as a result of the disease.

## 4. Discussion

This is a systematic review of reported cases of primary liver PEComas. These neoplasms are more frequently observed in females and affect individuals across a wide age range, though they are more commonly diagnosed in middle-aged patients [15]. Cases have been reported globally, with contributions from various regions. Most articles included in our analysis originated from Asia. Our data are consistent with another systematic review on hepatic PEComas by Kvietkauskas et al., in which 61.3% of reported cases were from Asian centers [16]. Possible explanations include a genetic predisposition in these populations for hepatic PEComa development and more effective reporting mechanisms in these countries.

The pathophysiology of these tumors is not fully understood, and the normal cell of origin for PEComas has not been identified. Hypotheses include the abnormal differentiation of smooth muscle progenitor cells found in concurrent malignancies such as leiomyosarcomas [17] and the simultaneous expression of smooth muscle and melanocytic phenotypes by dysregulated multipotent stem cells of neural crest origin [18]. PEComas have also been observed in patients with genetic disorders. TSC is an autosomal dominant genetic condition characterized by the presence of tumors in many different body organs and neuropsychiatric disorders [19]. Mutations of the TSC1 and TSC2 genes can affect the mechanistic target of the rapamycin (mTOR) signaling pathway and thus disrupt the regulating mechanisms of cell growth and proliferation, leading to tumor development [20]. However, the dysregulation of the mTOR pathway has also been observed in sporadic hepatic PEComas without TSC evidence. This observation has made the mTOR pathway a suitable treatment target for PEComas even in patients without TSC [21]. Rearrangements of the transcription factor E3 (TFE3) gene are another genetic abnormality associated with these patients [22]. These cases often have different clinical and morphological characteristics compared to conventional PEComas. They are typically found in younger patients without a history of TSC, have a distinct alveolar morphology, exhibit strong TFE3 immunohistochemical reactivity, and are usually not reactive to smooth muscle markers [23]. These findings have led to the suggestion that these tumors might represent a clinical condition with different molecular characteristics [24]. In our study, some liver PEComas were found in patients with Li–Fraumeni syndrome. Li–Fraumeni syndrome is an autosomal dominant condition where mutations in the tumor suppressor protein P53 gene (TP53) contribute to cancer development [25]. TP53 germline mutations have been suggested as a driver of PEComa development in recent years, especially in patients where other notable gene mutations are not found. These tumors often exhibit an aggressive malignant behavior, and concurrent malignancies in many different body areas might occur in these cases [26,27]. Despite their association with PEComa development, these mutations have been found only in a minority of patients, making the pathogenesis of these neoplasms a subject for further investigation.

Most patients with liver PEComas do not have a history of prior liver disease. Many tumors are asymptomatic and are discovered incidentally during routine physical examinations or imaging. Symptoms, when present, are non-specific and may include abdominal pain or discomfort, nausea, vomiting, or fever [28]. Laboratory tests are also non-specific, and markers such as AFP are mainly useful in ruling out other diseases in the differential diagnosis. Liver PEComas typically present as single lesions, although a multinodular appearance does not exclude the diagnosis. They can occur in various liver locations, but they are more frequently found in the right lobe [29]. This may be because the right lobe receives the largest portion of portal vein flow and is thus more exposed to carcinogens and other harmful stimuli from the GI tract [30]. However, the fact that many PEComas occur in other liver regions suggests the possibility of additional developmental mechanisms.

The most common type of PEComa is AML. These tumors often contain a high concentration of mature adipose tissue and centrally located, thick-walled, feeding blood vessels. These features often make them identifiable through routine imaging studies. However, many cases lack these characteristics, complicating their differentiation from other liver tumors [31]. Various histologic subtypes of AML, such as conventional, inflammatory, and epithelioid AMLs (eAMLs), have been described. Conventional AMLs can exhibit lipomatous, myomatous, or angiomatous predominance and are generally benign. eAMLs, by contrast, are more frequently associated with malignant transformation, and closer follow-up is advised for these patients [32].

Hepatic PEComas lack specific radiological characteristics. On imaging, hepatic PEComas are often enhanced in the arterial phase after contrast administration and exhibit variable washout patterns [33]. However, these neoplasms can present with many different imaging features and are often misdiagnosed as different liver tumors. In the present study, PEComa enhancement was observed in the arterial phase in one out of three patients after contrast administration, with a subsequent washout sign in the delayed or portal phases. These observations often make their distinction from HCC very difficult, especially in patients with underlying liver disease and cirrhosis. However, low aFP levels and normal liver parenchyma should alert clinicians to the possibility of hepatic PEComa presence [34]. The well-defined margins and high vascularity of PEComas can also make radiologists confuse them for hepatic hemangiomas [35]. The use of contrast-enhanced imaging can help in these cases. Hemangiomas exhibit a notable peripheral enhancement in the arterial phase, which is usually followed by centripetal fill-in in later phases [36]. FNH is another important mimic of liver PEComas. Both these modalities appear more often in young women and show arterial enhancement after contrast administration. The presence of a large feeding artery and a central scar in CT and MRI imaging is typical of FNH and can make its identification clear [37].

In our study, only 10.3% of PEComas were correctly identified via radiology. Several studies have focused on describing PEComa characteristics across different imaging modalities [38,39]. The presence of fat can aid in their identification, particularly in AML cases. On ultrasound, PEComas typically appear hypoechoic or isoechoic relative to liver parenchyma, and Doppler imaging may reveal a rich vascular supply. While not always observed, the presence of the washout sign on contrast-enhanced ultrasound (CEUS), CT, or MRI can complicate the differential diagnosis with HCC [29]. Despite these challenges, PEComas should be considered in cases of solitary liver lesions in patients without cirrhosis and with negative AFP, even if the washout sign is present. Ma et al. recently proposed that the appearance of an early draining vein and a more significant reduction in T1 relaxation time during the hepatobiliary phase of gadolinium-enhanced abdominal MRI may help distinguish PEComas from HCC [40]. PET-CT scans can help rule out metastatic disease and assess treatment response, but their use is limited by availability and the variable fluorodeoxyglucose uptake of PEComas [41].

Most liver PEComas are diagnosed postoperatively through pathological analysis. As PEComas mimic HCC, they are often initially presumed to be malignant, leading to more extensive liver resections such as segmentectomy or lobectomy. Biopsies can pose risks, including bleeding and malignant spread through the needle tract, potentially causing tumor recurrence [42]. Consequently, many clinicians avoid preoperative biopsies in lesions suspected to be HCC, relying instead on imaging for diagnosis. Moreover, diagnosis of PEComa via biopsy is not always definitive, making its routine use controversial [27]. Nevertheless, ultrasound-guided biopsies can support accurate tumor identification through early immunohistochemical analysis, aid in preoperative planning, and should be considered when malignancy is uncertain [9]. Pathological analysis with immunohistochemistry remains the most reliable diagnostic method for PEComas. The characteristic epithelioid cells typically express melanocytic markers such as HMB-45 and MELAN-A, along with smooth muscle markers like SMA. Due to their mesenchymal origin, they may also express markers such as vimentin [43]. In the liver, additional markers, such as HepPar-1 and AFP for hepatocytes, and cytokeratins and EMA for cholangiocytes, can further assist in diagnosis [44]. Despite its utility, some PEComas may evade detection even after pathological examination [45], emphasizing the need for increased awareness among clinicians and pathologists regarding these tumors.

Surgery is the preferred treatment for PEComas. Complete tumor resection with clear margins can be curative, with low recurrence rates reported [46]. A conservative approach involving close observation with laboratory and imaging follow-up has also been employed in some cases of benign PEComas [47]. An initial non-surgical approach was taken in 30 patients in our study, of whom 13 (43.3%) eventually required surgery due to tumor progression. As most hepatic PEComas are benign, a non-surgical approach is an option for clinicians, especially in cases with small tumor size, lack of malignant characteristics, and contraindications to surgery. However, these patients must be closely monitored due to the possible enlargement and malignant transformation of PEComas [9]. In inoperable cases due to tumor size or advanced stage, neoadjuvant treatments should be considered. Stereotactic radiation therapy and chemotherapy have shown success in reducing tumor size and vascularity [48,49]. Liver transplantation is another option for selected patients, while TACE and mTOR inhibitors have also been used for tumor reduction prior to surgery [50]. mTOR inhibitors have been increasingly used for the treatment of PEComas in recent years, either as a first-line treatment in inoperable malignant cases or as adjuvant therapy after surgery [51]. The identification of increased mTOR pathway activity even in PEComa cases without TSC mutations has made this protein a reasonable therapeutic target. In a phase II trial of patients with malignant PEComas, the effect of the intravenous administration of *nab*-sirolimus, an mTOR inhibitor, was examined. Patients exhibited a rapid response to treatment with favorable disease control rates, while severe adverse events were not observed [52]. These agents have also had success as a neoadjuvant treatment in cases of hepatic PEComas, where their use contributed to tumor shrinkage and made surgical treatment of these patients possible [21,53]. Further studies are required to identify the optimal dose for liver PEComa treatment, with which patients can benefit the most from mTOR inhibition.

Most liver PEComas are benign, but malignant forms have been reported. Due to their malignant potential and the risks associated with tumor growth, close monitoring is recommended, as surgical intervention may ultimately be necessary [35]. Vascular invasion and complications such as Budd–Chiari syndrome and portal vein thrombosis are associated with poor prognosis [54]. Metastasis can occur in the liver or distant organs, even years after initial resection [55]. Additional treatments such as TACE, TAE, and targeted agents like sorafenib can be beneficial in these cases. mTOR inhibitors have shown promise in improving outcomes for patients with advanced or metastatic PEComas [56,57]. Early pathological recognition of malignant features can enhance patient management by identifying those requiring close surveillance, even after successful surgery. Folpe et al. proposed a classification system for PEComas based on the presence of two or more high-risk features observed during pathological analysis. These include tumor size larger than 5 cm, an infiltrative growth pattern, a high nuclear grade, high cellularity, necrosis, a mitotic rate higher than 1 mitosis/50 high-power fields, and vascular invasion. A large tumor or nuclear polymorphism alone categorizes the tumor as having uncertain malignant potential [58]. Many researchers have used these criteria for classification in recent years. More recently, Yoo et al. proposed specific risk factors for primary hepatic PEComas, which demonstrated good predictive value. Worrisome features included a tumor size ≥ 7 cm, infiltrative borders, a mitotic rate > 1/10 mm^2^, necrosis, vascular invasion, and a diagnosis of PEComa-NOS. Patients with three or more of these features were classified as high-risk, while those with none were considered low-risk [59]. Applying these criteria in future pathological assessments may help predict prognosis and guide treatment strategies.

Our study has certain limitations. Due to the large number of case reports and series, the data included were heterogeneous in terms of design, population, and interventions, precluding meta-analysis. Many articles were excluded because they were written in non-English languages or lacked follow-up data necessary to evaluate outcomes. Nonetheless, these excluded studies may have contained valuable information regarding the radiological and pathological characteristics of liver PEComas that were not utilized in this review. The significance of this systematic review lies in its comprehensive description of primary liver PEComa characteristics across a substantial number of adult patients from diverse geographic regions. The volume of cases included provides valuable insights that can assist clinical decision-making and support the development of formal guidelines for the evaluation and treatment of primary hepatic PEComas. This study also underscores the importance of physician awareness and highlights the value of a multidisciplinary approach involving various medical specialties to determine the most effective management strategies.

## 5. Conclusions

Primary hepatic PEComa is a rare mesenchymal liver tumor with mostly benign clinical behavior, but variable malignant potential. It presents more frequently in women and is a solid liver lesion without specific clinical or radiological characteristics in the majority of cases, leading to frequent misdiagnosis. Although many cases are associated with genetic disorders such as TSC and Li–Fraumeni syndrome, several cases are sporadic. Pathological analysis remains the gold standard for diagnosis, but the evolution of imaging studies and increased awareness of the disease by many medical specialties can contribute to earlier PEComa recognition and optimal treatment. The recognition of high-risk patients through the use of specific risk stratification criteria based on worrisome features can identify patients in need of close follow-up and guide clinical decision-making. Such criteria are not currently established, and future controlled studies are required for their development.

Although surgical resection remains the cornerstone of management, prospective studies examining optimal patient treatment and surveillance strategies are currently lacking. Moreover, the expanding use of mTOR inhibitors and other possible targeted therapies offers additional options for the treatment of these patients, especially in malignant or inoperable cases. Many liver PEComas exhibit different biological behavior, especially in cases of mutations such as TFE3 rearrangement. Future studies are needed to determine optimal treatment choices and surveillance intervals based on specific tumor and patient characteristics. The establishment of formal evidence-based guidelines regarding the treatment of liver PEComas can support clinician decisions regarding this entity and further improve patient care.

## Figures and Tables

**Figure 1 cancers-17-02276-f001:**
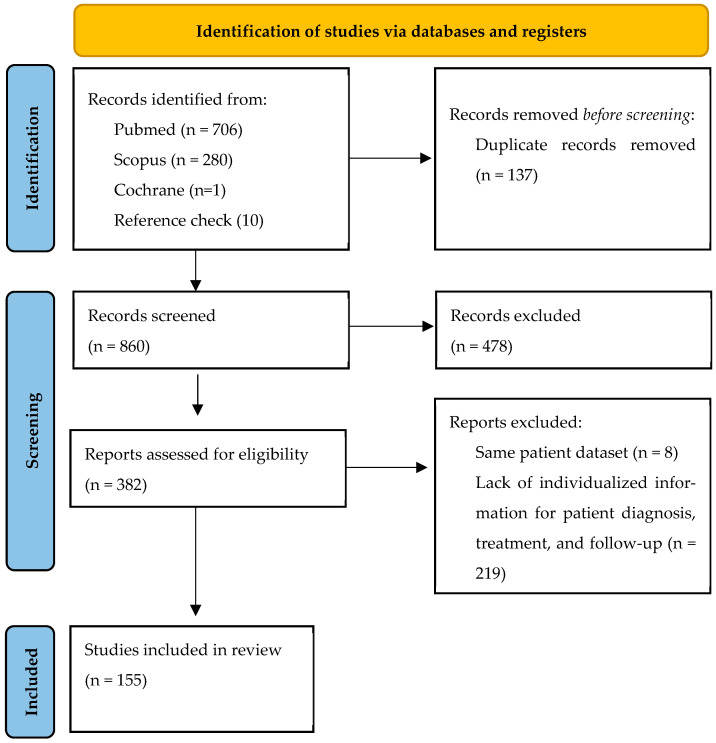
Flow chart of study selection.

**Table 1 cancers-17-02276-t001:** Patient demographics, medical history, and tumor features in reported cases of primary liver PEComas.

Parameter	Value
Study origin	
Asia	97 studies (66.6%)
Europe	35 studies (24.1%)
North America	8 studies (5.5%)
South America	3 studies (2.1%)
Africa	1 study (0.7%)
Australia	1 study (0.7%)
Age	46 years (IQR: 35.25–53.75)
Females	208 patients (74%)
Comorbidities	
Hepatitis B	23 patients (8.2%)
Hepatitis C	5 patients (1.8%)
History of extrahepatic malignancy	30 patients (10.7%)
TSC	9 patients (3.2%)
Li–Fraumeni syndrome	6 patients (2.1%)
Presentation	
Asymptomatic	157 patients (55.9%)
Abdominal pain	84 patients (29.9%)
Abdominal discomfort	24 patients (8.5%)
Fever	15 patients (5.3%)
Weight loss	10 patients (3.6%)
Fatigue	7 patients (2.5%)
Vomiting	5 patients (1.8%)
Nausea	5 patients (1.8%).
Diagnostic workup	
U/S	152 patients (54.1%)
CT	221 patients (78.6%)
MRI	156 patients (55.5%)
PET-CT	31 patients (11%)
Tumor size	5.15 cm (IQR: 3–9.3)
Tumor location	
Right lobe	149 cases (53%)
Left lobe	101 cases (34%)
Caudate lobe	12 cases (4.3%)
Tumor nodules	
Single liver lesion	249 patients (88.6%)
Multiple nodules	22 patients (7.8%)
Positive immunohistochemical markers	
HMB-45	261/264 (98.9%)
MELAN-A	124/132 (93.2%)
SMA	179/200 (89.5%)
Vimentin	62/79 (78.5%)
Negative immunohistochemical markers	
S-100	85/126 (67.5%)
Desmin	45/83 (54.2%)
Cytokeratins	123/125 (98.1%)
EMA	26/28 (92.9%)
HepPar-1	43/44 (97.7%)
Final diagnosis	
Radiology	29 cases (10.3%)
Biopsy	44 cases (15.7%)
Post-surgical analysis	208 cases (74%)
PEComa type	
AML	197 patients (70.1%)
PEComa-NOS	74 patients (26.3%)
CCMMT of the falciform ligament	7 patients (2.5%)
Clear-cell sugar tumor	3 cases (1.1%)
Malignancy	30 cases (10.7%)
Surgery	251 patients (89.3%)
Follow-up duration (months)	24 months (IQR: 12–48)
Recurrence	17 patients (6%)
Death due to PEComa	8 patients (2.85%)

Abbreviations: IQR, Interquartile Range; TSC, Tuberous Sclerosis Complex; U/S, Ultrasound; CT, Computed Tomography; MRI, Magnetic Resonance Imaging; PET-CT, Positron Emission Tomography–Computed Tomography; HMB-45, Human Melanoma Black-45; MELAN-A, Melanoma Antigen Recognized by T cells A; SMA, Smooth Muscle Actin; S-100, S-100 protein; EMA, Epithelial Membrane Antigen; HepPar-1, Hepatocyte Paraffin 1; AML, Angiomyolipoma; PEComa, Perivascular Epithelioid Cell Tumor; PEComa-NOS, Perivascular Epithelioid Cell Tumor—Not Otherwise Specified; CCMMT, Clear-Cell Myomelanocytic Tumor.

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
