# Peer review of "Characteristics and Treatment of Primary Hepatic Perivascular Epithelioid Cell Tumor (PEComa) in Adults: A Systematic Review"

_cancers, 2025, doi:10.3390/cancers17142276_

Round 1
Reviewer 1 Report
Comments and Suggestions for Authors
Dear Authors,
Thank you for submitting your manuscript to Cancers. After careful consideration, we feel that it has merit but does not fully meet Cancers publication criteria as it currently stands. The shortcomings of this paper needs to be worked out before it can be considered for publication. Therefore, we invite you to resubmit a revised version of the manuscript that addresses the points raised during the review process.
The manuscript titled Characteristics and Treatment of Primary Hepatic Perivascular Epithelioid Cell Tumor (PEComa) in adults: a Systematic Review” demonstrated that the
Primary hepatic PEComa is a rare liver tumour with mostly benign clinical behaviour and nonspecific presentation. It presents more frequently in women and is a solid liver lesion in the majority of cases. The recognition of high risk patients through the use of risk stratification criteria based on worrisome features can identify patients in need of close follow up and guide clinical decision-making.
- The Authors should briefly introduce the known molecular drivers of PEComas and their relevance to hepatic tumors link between PEComas and TSC and also the pathogenesis, behavior of malignant PEComas, and potential molecular features.
- The Authors must provide examples of how hepatic PEComas are misdiagnosed or mimic other liver tumors (e.g., HCC, hemangioma, focal nodular hyperplasia).
- The Authors should also mention why a systematic review is important.
- The manuscript seems to be incomplete and it could have been much better with more information and proper proposed molecular mechanistic pathways with a suitable figure.
- The conclusions cannot be justified on the basis of the rest of the paper. The conclusion succinctly summarizes the findings but could be expanded to more explicitly state the potential clinical implications of your research and suggest specific areas for future studies. The authors must rewrite the conclusion part.
- Streamline the general discussions to ensure they are concise and directly relevant. Focus on key aspects that directly support or relate to the main theme of the manuscript.
- Explore the potential of targeted therapies (e.g., mTOR inhibitors) and compare surgical outcomes with non-surgical approaches, if reported. Present more comparative analysis between benign and malignant cases if possible.
- The authors should include a PRISMA flow diagram to justify their manuscript.
- The literature review lacks depth and does not reflect the most recent advancements in the field. While it references some relevant studies, it overlooks several key contributions that are critical to the topic. Please review and update the references to include current and significant research, particularly recent developments that could enhance the context and relevance of your discussion.
- Several grammatical errors and awkward phrasings are present throughout the document which could hinder its readability and professional presentation. Therefore, I recommend the paper should undergo professional language editing before it can be published.
- Several grammatical errors and awkward phrasings are present throughout the document which could hinder its readability and professional presentation. Therefore, I recommend the paper should undergo professional language editing before it can be published.
Author Response
Thank you for taking the time to help us improve our manuscript. Here is our response to your comments:
1. The Authors should briefly introduce the known molecular drivers of PEComas and their relevance to hepatic tumors link between PEComas and TSC and also the pathogenesis, behavior of malignant PEComas, and potential molecular features.
Response 1: The known molecular drivers of PEComas have been introduced. Information regarding RSC, Li Fraumeni syndrome and TFE3 rearrangements have been added and molecular features of these tumors are included in lines 234-258 and are highlighted in blue. We have added and cited references accordingly.
2. The Authors must provide examples of how hepatic PEComas are misdiagnosed or mimic other liver tumors (e.g., HCC, hemangioma, focal nodular hyperplasia).
Response 2: A paragraph has been added in the Discussion section, lines 280-297, which describes how PEComas might mimic other liver tumors and possible features that could help in their differentiation. This paragraph is highlighted in yellow.
3. The Authors should also mention why a systematic review is important.
Response 3: The importance of a systematic review regarding this subject has been stressed in the Introduction section, in lines 59-74. Added information is highlighted in yellow.
4. The manuscript seems to be incomplete and it could have been much better with more information and proper proposed molecular mechanistic pathways with a suitable figure.
Response 4: More information regarding the molecular pathways and pathogenesis of liver PEComas have been added to the text, as was mentioned in Response 1. We have added and cited references accordingly. While we acknowledge the value of including a visual representation, we chose not to add a figure illustrating the proposed molecular mechanisms, as our primary aim was to highlight the clinical features and diagnostic challenges of these rare tumors. We believe that an extensive focus on molecular mechanisms might shift the emphasis away from the clinical perspective, which constitutes the core of our systematic review.
5. The conclusions cannot be justified on the basis of the rest of the paper. The conclusion succinctly summarizes the findings but could be expanded to more explicitly state the potential clinical implications of your research and suggest specific areas for future studies. The authors must rewrite the conclusion part.
Response 5: The Conclusions section has been rewritten to more explicitly state the potential clinical implications of our research and suggest specific areas for future studies. These changes can be found in lines 395-416 highlighted in yellow.
6. Streamline the general discussions to ensure they are concise and directly relevant. Focus on key aspects that directly support or relate to the main theme of the manuscript.
Response 6: We believe that the information and references we have added regarding hepatic PEComa pathophysiology, differential diagnosis and recent treatment advances further support the main theme of the manuscript.
7. Explore the potential of targeted therapies (e.g., mTOR inhibitors) and compare surgical outcomes with non-surgical approaches, if reported. Present more comparative analysis between benign and malignant cases if possible.
Response 7: The potential of targeted mTOR inhibitor therapy has been analyzed in the discussion section in lines 345-356. These lines are highlighted in blue. We have added and cited references accordingly. As meta-analysis was not done, a direct comparison of outcomes in different patient groups was not possible. However, such comparisons are important for the evolution of our understanding and treatment of these tumors and so we suggest that such studies should take place in the future in the Conclusions section.
8. The authors should include a PRISMA flow diagram to justify their manuscript.
Response 8: A PRISMA flow diagram is included in the manuscript and is shown in Figure 1, while it is also provided as a separate file during submission.
9. The literature review lacks depth and does not reflect the most recent advancements in the field. While it references some relevant studies, it overlooks several key contributions that are critical to the topic. Please review and update the references to include current and significant research, particularly recent developments that could enhance the context and relevance of your discussion.
Response 9: We believe that the information and references we have added in this revised version enhance the context and relevance of the discussion section.
10. Several grammatical errors and awkward phrasings are present throughout the document which could hinder its readability and professional presentation. Therefore, I recommend the paper should undergo professional language editing before it can be published.
Response 10: Several changes regarding grammar and vocabulary have been made in the text after an evaluation by a native English speaker. We hope you will find the revised version of the manuscript much improved.
Reviewer 2 Report
Comments and Suggestions for Authors
This is a well-executed and clinically valuable review on a rare hepatic neoplasm.
I have only two minor suggestions to improve the quality of the manuscript:-
- In Figure 1 it is not clear to me what the "*" and "**" stand for. In the figure legend there are no references for these symbols
- Table 1 is too bulky. Could it be submitted as supplementary material?
Author Response
Thank you for taking the time to help us improve our manuscript. Here is our response to your comments:
- In Figure 1 it is not clear to me what the "*" and "**" stand for. In the figure legend there are no references for these symbols
Response: These symbols have been removed from Figure 1. We apologize for any confusion.
- Table 1 is too bulky. Could it be submitted as supplementary material?
Response: Table 1 has been removed from the main text and resubmitted as supplementary table S1 in a separate file. Table 2 has been renamed and is now Table 1.
Reviewer 3 Report
Comments and Suggestions for Authors
The authors performed a systematic review of reported case reports and case series of primary liver PEComas to determine patient baseline characteristics, clinical course, pathological, immunohistochemical, and radiological tumor features, as well as the method and timing of diagnosis and treatment types. This review could be informative for readers. However, this reviewer has some comments.
Comments:
- In Table 1, all the cases are listed, but it had better be transferred it to a supplementary table, and the number of cases in each category, such as continent, sex, and others, should be shown in the main text.
- It is noteworthy that PEComas have been observed in patients with genetic disorders such as TSC and Li-Fraumeni syndrome. More information about the genetics of PEComas is expected to be shown.
- Although the diagnosis of PEComas is well discussed, the treatment of PEComas seems not fully discussed.
Author Response
Thank you for taking the time to help us improve our manuscript. Here is our response to your comments:
1. In Table 1, all the cases are listed, but it had better be transferred it to a supplementary table, and the number of cases in each category, such as continent, sex, and others, should be shown in the main text.
Response 1: Table 1 has been removed from the main text and resubmitted as supplementary table S1 in a separate file. Table 2 has been renamed Table 1. The number of cases in each category is mentioned in the main text.
2. It is noteworthy that PEComas have been observed in patients with genetic disorders such as TSC and Li-Fraumeni syndrome. More information about the genetics of PEComas is expected to be shown.
Response 2: The known molecular drivers of PEComas have been introduced. Information regarding RSC, Li Fraumeni syndrome and TFE3 rearrangents have been added and molecular features of these tumors are included in lines 234-258 and are highlighted in blue. We have added and cited references accordingly.
3. Although the diagnosis of PEComas is well discussed, the treatment of PEComas seems not fully discussed.
Response 3: We have added more information regarding the treatment of PEComas in the Discussion section, in lines 335-341 and 345-356. These lines are highlighted in blue.
Round 2
Reviewer 1 Report
Comments and Suggestions for Authors
The authors have satisfactorily addressed most of my concerns in a thorough and satisfactory fashion. I consider the manuscript acceptable for publication.
Comments on the Quality of English LanguageStill some typos are there which could have been corrected before final submission.
Reviewer 3 Report
Comments and Suggestions for Authors
All the comments have been addressed.